# The prevalence and determinants of vitamin D deficiency in Indonesian infants at birth and six months of age

Vicka Oktaria[1,2,3]*, Stephen M. Graham[1], Rina Triasih[2,3], Yati Soenarto[2,3], Julie E. Bines[1], Anne-Louise Ponsonby[1], Michael W. Clarke[4,5], Rizka Dinari[3], Hera Nirwati[3,6], Margaret Danchin[1]

1 Department of Paediatrics and Murdoch Childrens Research Institute, Royal Children's Hospital, University of Melbourne, Melbourne, Victoria, Australia, 2 Child Health Department, Faculty of Medicine, Public Health and Nursing, Universitas Gadjah Mada, Yogyakarta, Indonesia, 3 Paediatrics Research Office, Child Health Department, Faculty of Medicine, Public Health and Nursing, Universitas Gadjah Mada, Yogyakarta, Indonesia, 4 Metabolomics Australia, Centre for Microscopy, Characterisation and Analysis, The University of Western Australia, Perth, Western Australia, Australia, 5 School of Biomedical Sciences, Faculty of Health and Medical Sciences, The University of Western Australia, Perth, Western Australia, Australia, 6 Microbiology Department, Faculty of Medicine, Public Health and Nursing, Universitas Gadjah Mada, Yogyakarta, Indonesia

* vicka.oktaria@ugm.ac.id

**Data Availability Statement:** All relevant data are within the manuscript and dataset is available on the Figshare repository (DOI 10.6084/m9.figshare.12931817).

## Abstract

### Background

Vitamin D deficiency in infants has been associated with an increased risk of a number of diseases but there are limited data on the prevalence and determinants of vitamin D deficiency from tropical settings with high infant morbidity and mortality.

### Objective

To determine the prevalence and determinants of vitamin D deficiency in infants at birth and at six months of age in Yogyakarta province, Indonesia.

### Design

Serum vitamin D of eligible infants was measured in cord blood at birth and at six months of age. Factors associated with vitamin D deficiency (serum 25-hydroxyvitamin D <50 nmol/L) were collected prospectively monthly from birth and concentrations measured by liquid chromatography-tandem mass spectrometry. Independent risk factors were identified by multiple logistic regression.

### Results

Between December 2015 to December 2017, 350 maternal-newborn participants were recruited and followed up. Vitamin D deficiency was detected in 90% (308/344) of cord blood samples and 13% (33/255) of venous blood samples at six months. Longer time outdoors (≥2 hours per day) and maternal multivitamin intake containing vitamin D during pregnancy were protective against vitamin D deficiency at birth (AOR: 0.10, 95% CI: 0.01–0.90 and AOR: 0.21,

**Funding:** This study was supported by Murdoch Children's Research Institute in the form of funding awarded to SMG, Schlumberger foundation faculty for the future in the form of funding awarded to VO, Indonesia Endowment Fund for Education (LPDP) Ministry of Finance in the form of a grant awarded to VO (20130822080370), the David Bickart Clinician Research Fellowship from the University of Melbourne awarded to MD, Australia-Indonesia Centre (AIC) in the form of a grant awarded to MD and YS (01HSP1MELDancUGM003), and infrastructure funding from the Western Australian State Government, in partnership with the Australian Federal Government, through Bioplatforms Australia and the National Collaborative Research Infrastructure Strategy awarded to MWC. The funders had no role in study design, data collection and analysis, decision to publish, or preparation of the manuscript.

**Competing interests:** The authors have declared that no competing interests exist.

**Abbreviations:** EBF, Exclusive breastfeeding; IPAD study, Indonesian Pneumonia and Vitamin D study; IQR, Interquartile range; LC-MS/MS, Liquid chromatography-tandem mass spectrometry; OR, Odds ratio; AOR, Adjusted odds ratio; REDCap, Research Electronic Data Capture; SD, Standard deviation; UV, Ultraviolet; UVB, Ultraviolet B; UVR, Ultraviolet radiation; 25(OH)D, 25-hydroxyvitamin D.

95% CI: 0.06–0.68, respectively). Risk factors for vitamin D deficiency at six months included lower cumulative skin-sun exposure score (AOR: 1.12, 95% CI: 1.04–1.20), severe vitamin D deficiency at birth (AOR: 7.73, 95% CI: 1.20–49.60) and exclusive breastfeeding (AOR: 2.64, 95% CI: 1.07–6.49) until six months. Among exclusively breast fed (EBF) infants, a higher skin-sun exposure score was associated with reduced vitamin D deficiency risk.

## Conclusion

In equatorial regions, the role of 'safe' morning sun exposure in infants and mothers in populations with medium to dark brown skin pigmentation and effective interventions to prevent vitamin D deficiency in newborns and EBF infants, need further consideration and evaluation.

## Introduction

Over recent decades, there has been a rapid growth of research interest in perinatal vitamin D deficiency due to a range of potential roles of vitamin D for prevention of disease, such as low birth weight in newborns or infections in early childhood [1–5]. A temperate country such as Australia with a high prevalence of infant vitamin D deficiency, particularly in winter, has implemented preventive measures through routine vitamin D supplementation for high-risk pregnant women and infants (i.e. those with dark skin type and covering clothing), in addition to guidance around safe sun exposure practices [6]. Despite abundant sunlight, vitamin D deficiency is also common in tropical countries, but under-diagnosed. Recent studies have reported that one in every two African and nine in every ten Thai neonates were vitamin D deficient [7, 8]. In tropical regions that experience high neonatal and child health morbidity and mortality, such as in Africa and Southeast Asia [9, 10], sustaining adequate vitamin D level could be a low-cost public health measure to reduce the burden of infectious diseases. However, efforts to prevent, detect, and treat vitamin D deficiency in many of these regions are minimal or absent.

Understanding the epidemiology and identification of risk factors for vitamin D deficiency in early life is important for future development of preventive strategies but they are not currently well understood in Indonesia. Indonesia is the most populous country in Southeast Asia and has the fifth-highest global burden of infant morbidity and mortality from acute respiratory infections [11, 12]. Furthermore, little is also known about cultural practices that may affect vitamin D status in the Indonesian population. One particular practice of interest is infant sunbathing in the postnatal period [13]. Infant sunbathing practices have been routine among Indonesian mothers for decades and are believed to improve neonatal jaundice, a common condition in Indonesian newborns [14, 15], as well as be an important source of vitamin D [16, 17]. Excessive sun exposure is a risk factor for skin malignancies and needs to be balanced against the beneficial effects for vitamin D, considering skin type, sun behavior, and atmospheric conditions [18]

This study aimed to determine the prevalence of vitamin D deficiency in infants at birth and at six months and to explore the environmental, social, and maternal behavioral factors during pregnancy associated with vitamin D deficiency at birth and behavioral factors associated with vitamin D deficiency at six months.

## Materials and methods

This study was part of the Indonesian Pneumonia and Vitamin D status (IPAD study), a community-based birth cohort study conducted between December 2015 to December 2017 in nine Primary Healthcare Centers and five private clinics in Kota Yogyakarta (7.80˚ S, 110.4˚

E) and Kulon Progo regencies (7.83˚ S, 110.2˚ E), Yogyakarta province, Indonesia. Inclusion criteria included expected delivery to be in one of the study sites without family intention to leave the area within 12 months. Expectant mothers were approached in the third trimester to consent to the collection of 5 mL of cord blood at delivery and 2.5 mL of infant venous blood at six months of age at the Primary Healthcare Center or private clinic for vitamin D testing. Newborns were recruited at birth and followed monthly for 12 months after written consent had been obtained from both parents.

Soon after birth, mothers completed a structured questionnaire interview on their diet, time spent outdoors exposed to direct sunlight, the area of skin that was exposed to direct sunlight, and any sun protection used during pregnancy. Similar data on infant sun exposure and behavior as well as infant feeding practices were collected prospectively every month by face-to-face or phone interviews. The mother's and infant's skin types were determined at the first face-to-face visit for mothers and at the six-month visit for infants with the Fitzpatrick visual skin chart, a reliable instrument if spectophotometry is not available [19]. Skin type was measured in the area that was least exposed to sunlight, such as in the inner arms [19].

## Study assays and measurement for vitamin D

We measured total serum 25-hydroxyvitamin D3 and 25-hydroxyvitamin D2 concentrations using liquid chromatography-tandem mass spectrometry (LC-MS/MS), the current gold standard for vitamin D measurement. The LC-MS/MS analysis was performed by Metabolomics Australia, at the University of Western Australia. The analytical sensitivity (or limit of detection) and the functional sensitivity (or limit of quantitation) of the assay for 25-hydroxyvitamin D3 is 0.5 nM and 2 nmol/L respectively. The center has been certified by the Center for Disease Control and Prevention for standardized vitamin D measurement ($r^2$ = 0.99) [20]. Vitamin D deficiency was defined as a concentration of 25-hydroxyvitamin D of less than 50 nmol/L. The vitamin D concentrations at birth and at six months of age were also categorized by severity using the categories suggested by Paxton *et al.*: mild deficiency (30–49 nmol/L); moderate deficiency (12.5–29 nmol/L) and severe deficiency (<12.5 nmol/L) [6].

## Data acquisition for Ultraviolet B

Ultraviolet B (UVB) and Erythema Daily Dose Rate data for the Yogyakarta Special Region were downloaded monthly and obtained from the Aura Data Validation Centre as a part of the National Aeronautics and Space Administration, Goddard Space Flight Centre, Greenbelt, Maryland. (URL link: https://avdc.gsfc.nasa.gov/index.php?site=2057856112&id=79)

## Ethics

We obtained ethical approval for the study from the University of Melbourne Human Research Committee (ethics approval number 1544817, October 2015) and the Medical and Health Research Ethics Committee of Universitas Gadjah Mada, Yogyakarta, Indonesia (ethics approval number KE/FK/935/EC/2015, July 2015).

## Study definitions

Composite estimate UVR exposure and skin-sun exposure/protection score were measured in mother and infant. Composite estimate UVR exposure dose variables were created by multiplying the time spent outdoors and the ambient UVB for the appropriate month and presented as a monthly or cumulative value [21]. Skin-sun exposure score was defined as the frequency of uncovered/exposed arms and legs (1 = never, 2 = less than 50% of the time, 3 = more than

50% of the time, 4 = all of the time) [21]. The score for skin exposure was the summation of each category (each month score: 2 = minimum sun exposure, 8 = maximum sun exposure; and 6 month cumulative score: 12 = minimum sun exposure, 48 = maximum sun exposure) [21]. Exclusive breastfeeding (EBF) infants were defined as infants who received only breast milk until six months of age.

### Data analysis

We entered de-identified data into REDCap database (Research Electronic Data Capture) and exported to STATA version 12 (Stata Corporation, College Station, Texas) for analysis. All variables were presented as mean ± SD (or median and interquartile ranges) and proportion (%), as appropriate. Correlation between cord blood and venous blood vitamin D was tested using the Pearson correlation test. Univariate and multivariate logistic regression was performed to explore the strength of the association between vitamin D status as a binary outcome and other categorical (i.e. monthly family income or exclusive breastfeeding status) or continuous variables (i.e. skin-sun exposure score or cumulative composite UV estimate) that served as risk factors for vitamin D deficiency at birth or at six months of age and are presented as odds ratios with 95% confidence intervals. Confounding was determined if the difference between the OR in the univariate and multivariate models changed by more than 10% after the inclusion of suspected confounding variables or if the variables have been commonly associated with the exposures or the outcomes in the existing literature. A p-value of <0.05 was considered statistically significant.

## Results

We recruited 350 participants to the study. Of these, 344 (98%) participants had cord blood samples, 255 (73%) participants provided a venous blood sample at six months of age and 249 (71%) participants had both cord and venous blood samples (Fig 1). The demographic characteristics for rural, urban, and all participants are presented in Table 1. Mothers in rural Kulon Progo were less educated, had lower family income and lower EBF rates, but household size (7 people) was greater in urban Kota Yogyakarta (21% vs 6%).

### Vitamin D status of Indonesian infants at birth and at six months of age

The prevalence of vitamin D deficiency was 90% in 344 participants at birth and 13% in 255 participants at six months (± 4 weeks) (Fig 2). The mean (± SD) of vitamin D concentrations at birth and six months were 30 (± 14) nmol/L and 77 (± 26) nmol/L respectively. There was a positive correlation between vitamin D concentration in cord blood and infants' venous blood at six months of age (Pearson's rho = 0.23, p-value = 0.03, Fig 3). Of those with vitamin D deficiency at birth, 87% (192/222) had normal vitamin D status by six months of age. Thirty-five percent (6/17) of participants who were severely deficient (<12.5 nmol /L) at birth had a vitamin D concentration below 50 nmol/L at six months of age (Table 2).

Seven percent of mothers (23/350) took a multivitamin containing vitamin D during pregnancy and 23% (78/350) of infants had formula feeding by the age of six months old (Table 3). Eighty four percent of the infants (280/334) sunbathed on a daily basis before 10 am with uncovered arms and legs before the age of six months.

### Factors associated with vitamin D status at birth

A longer time spent outdoors exposed to direct sunlight during pregnancy was a protective factor against vitamin D deficiency at birth (Table 4). Mothers who spent two hours or more

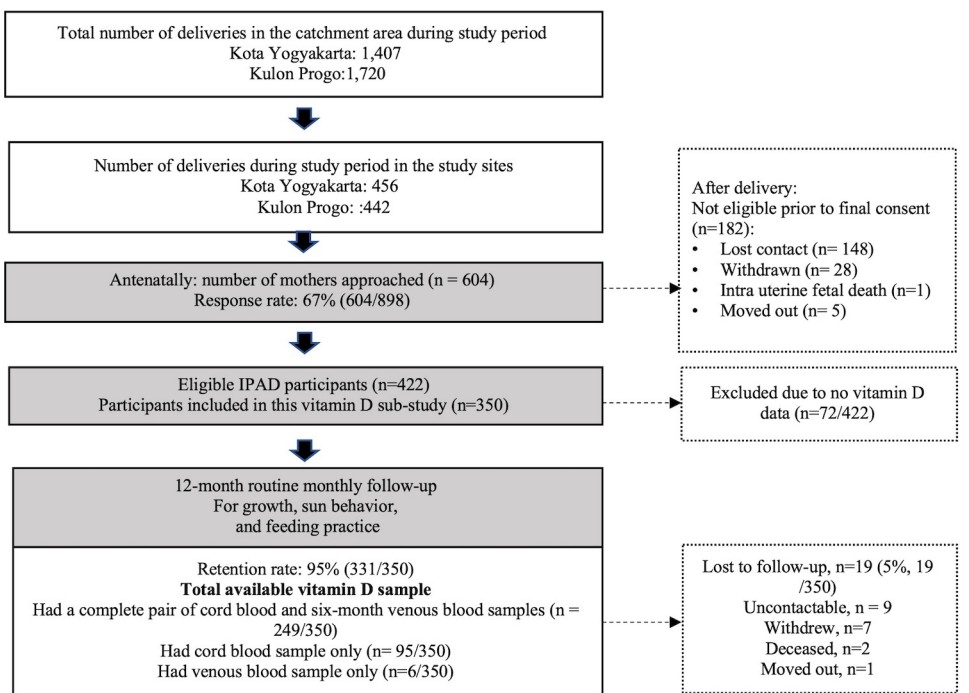

**Fig 1. Flow diagram depicts sample collection for vitamin D measurement and inclusion in the final analyses.**

per day outside were less likely to have a newborn with vitamin D deficiency at birth compared to those who spent only 15 minutes or less in the sun. Taking into account the monthly ambient UVB, by multiplying the ambient UVB with the average time spent outdoor during pregnancy, a higher cumulative composite estimate UVR exposure for nine months was protective against vitamin D deficiency at birth. Maternal intake of a multivitamin containing vitamin D during pregnancy was independently associated with a reduced risk of vitamin D deficiency at birth.

## Factors associated with vitamin D status at six months of age

Infants who were severely vitamin D deficient at birth were seven times more likely to be vitamin D deficient at six months of age (OR: 7.73, 95% CI 1.20–49.60) compared to those who were not deficient at birth (Table 5). EBF infants were twice as likely to have vitamin D deficiency at six months of age (OR: 2.64, 95% CI: 1.07–6.49) even after adjustment for sex, family income, skin type, and the month of blood collection. Lower skin-sun exposure score was an independent risk factor for vitamin D deficiency at six months of age (Table 5). In stratified analysis by EBF status, a higher magnitude of association between skin-sun exposure score and risk of vitamin D deficiency at six months of age was demonstrated only in EBF infants particularly on weekday exposure (AOR: 1.13,95% CI: 1.04–1.24, not shown in the table).

## Discussion

This study found a high prevalence of vitamin D deficiency in Indonesian infants at birth that decreased markedly by six months of age. Increased maternal and postnatal infant sun exposure reduced the risk of vitamin D deficiency at birth and six months respectively. Other risk factors for vitamin D deficiency at six months were severe vitamin D deficiency at birth and

**Table 1. Demographic characteristics of the eligible participants (N = 350)[1].**

| Variables | Kota Yogyakarta (Urban, N = 127) | Kulon Progo (Rural, N = 223) | All (N = 350) |
|---|---|---|---|
| Sex male–n (%) | 51 (40%) | 119 (54%) | 170 (49%) |
| Month of birth | | | |
| • January to March (Wet season) | 52 (41%) | 69 (31%) | 121 (35%) |
| • April to June (Dry season) | 47 (37%) | 86 (39%) | 133 (38%) |
| • July to September (Dry season) | 16 (13%) | 34 (15%) | 50 (14%) |
| • October to December (Wet season) | 12 (9%) | 34 (15%) | 46 (13%) |
| Birth weight in grams–(median, IQR), N = 348 | 3050 (2800–3300) | 3100 (2900–3400) | 3100 (2900–3350) |
| Mother's age in years–(median, IQR) | 28 (24–33) | 29 (24–33) | 29 (24–33) |
| Gestational age in weeks–(median, IQR), N = 349 | 40 (39–40) | 39 (39–40) | 40 (39–40) |
| Mother's educational level–n (%), N = 349 | | | |
| • Middle school or less | 23 (18%) | 60 (27%) | 83 (24%) |
| • High school | 89 (70%) | 137 (61%) | 224 (64%) |
| • University | 15 (12%) | 27 (12%) | 42 (12%) |
| Mother's occupation–n (%) | | | |
| • Stay home | 93 (73%) | 155 (70%) | 248 (71%) |
| • Working / student | 34 (27%) | 68 (30%) | 102 (29%) |
| Family income per month–n (%), N = 349 | | | |
| • < IDR 1000k | 55 (44%) | 122 (55%) | 177 (51%) |
| • IDR 1000k – 5000k | 67 (53%) | 93 (42%) | 160 (48%) |
| • IDR > 5000k | 4 (3%) | 8 (4%) | 12 (3%) |
| Number of people living with participant (median, IQR) | 5 (3–6) | 4 (3–5) | 4 (3–5) |
| 7 or more people living with participant–n (%) | 26 (21%) | 13 (6%) | 39 (11%) |

[1] N = 350, unless otherwise specified.

Indonesia is a tropical country with two seasons: the dry season (around April to October) and the wet season (around November to March).

EBF until six months of age. Among EBF infants, greater skin surface exposure to the sun was associated with a reduced risk of vitamin D deficiency.

High prevalence of vitamin D deficiency at birth has been previously documented with no clear pattern associated with latitude or ethnicity [22–24]. In Poland, New Zealand and the USA, the reported prevalence of vitamin D deficiency (<50 nmol/L) in cord blood was between 29% and 57%, particularly in low UV ambience seasons and dark-skinned people, while in Nigeria, the prevalence was 30% with an indoor lifestyle and maternal veils as the common risk factors [25–28]. We are not aware of previous studies that have evaluated vitamin D status in Indonesian newborns, though Yani et al. reported that of 168 Indonesian children aged less than five years old, one-third were vitamin D deficient [29]. Thai investigators recently reported a similar prevalence of cord blood vitamin D deficiency (89%) to our study but with a smaller sample size (N = 94) [8]. The limited data suggest that vitamin D deficiency in pregnant women may be common in South East Asian populations, as cord blood concentrations would reflect transplacental vitamin D supply [29, 30]. As such, there are recommendations for pregnant women relating to sun exposure in Indonesia, i.e. continuous sun exposure between 10 am to 1 pm for 30 minutes (or 60 minutes for veiled women) per day [31]. The findings here support the recommendations that pregnant women have adequate sun exposure, as higher sun exposure reduced the risk of vitamin D deficiency at birth.

There is a complex interplay between multiple factors, such as genetics, environment, and lifestyle that influence the epidemiology of vitamin D deficiency [32]. Despite a very high prevalence of vitamin D deficiency at birth, only a minority of infants in our study had persistent

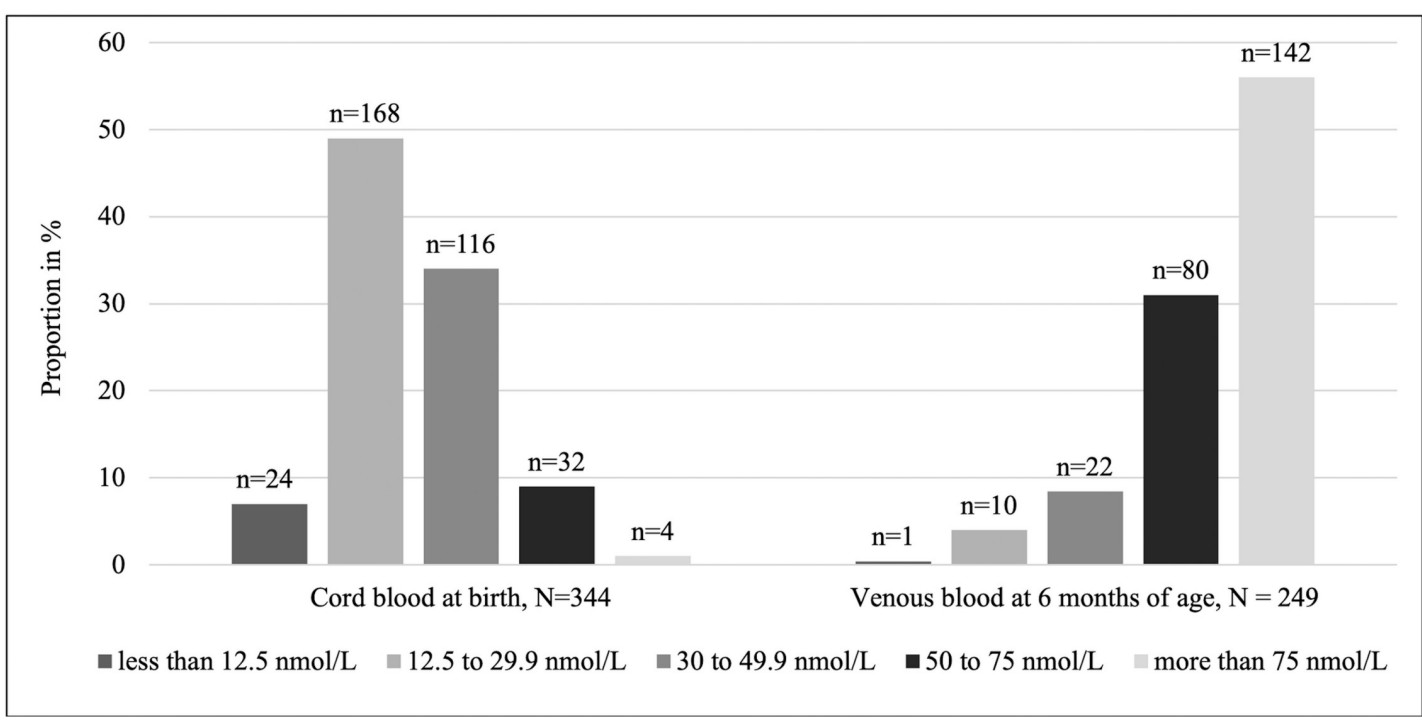

**Fig 2. The prevalence of vitamin D deficiency at birth and at 6 months of age.**

vitamin D deficiency at six months of age. Routine cultural practices of Indonesian mothers exposing their uncovered newborns directly to the morning sun may have contributed to cutaneous vitamin D production [14, 15]. Although we measured the infant skin-sun exposure score for consecutively for six months, using reflection on habitual sun behavior rather than temporal variations, our finding in favour of sun exposure needs careful interpretation as infant skin is vulnerable to sun damage [33, 34]. Experts highlight the need for a balance between endogenous vitamin D production and the risk of skin malignancy with respect to direct sunlight [32, 34, 35]. There is no universal agreement as to the sun-exposure required to be equivalent to the recommended dietary intake of at least 400 IU of vitamin D daily. To reach a minimal erythema dose or a slight pinkness in the skin (10,000–25,000 IU), brown to dark-skinned infants need five to ten times the duration of sun exposure than fair-skinned infants who need approximately 30 minutes per week of sun exposure wearing only a diaper [13, 34, 35]. The optimum time for cutaneous synthesis of vitamin D is between 10 am to 3 pm when the UV index is high, but the risk of skin cancer is increased [32]. Most of our infants were sunbathed with uncovered arms and legs before 10 am and their vitamin D concentrations improved by six months. This practice is perhaps sufficient for typical Indonesian skin and the environment but needs further confirmatory studies. Future research should determine the optimum duration and daytime for safe sun exposure for Indonesian infants.

Prenatal and postnatal vitamin D supplementation is not routinely recommended in our setting. Further, there are limited selections of food with a high content of vitamin D in the daily intake of Indonesian people [36]. Only 5% of our study infants had egg yolk in the first six months, mainly those who were living in the rural area, and even less had had fish or red meats. A minority (7%) of mothers reported taking a multivitamin supplement that contained vitamin D during pregnancy with 50% reporting a daily vitamin D intake below 400 IU, the minimum recommended daily dose in pregnancy [6]. None of the participating infants

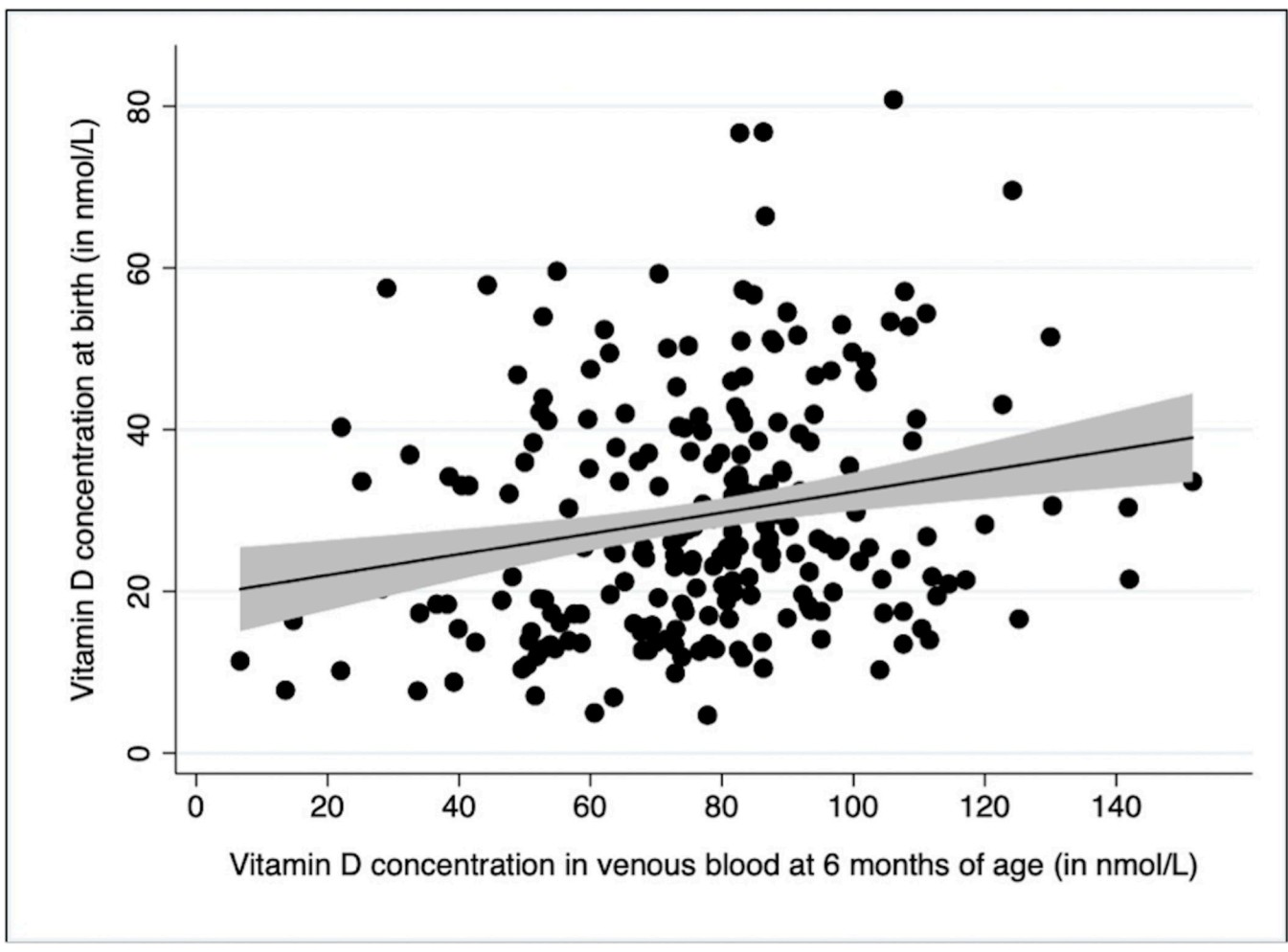

**Fig 3. Correlation between vitamin D concentration at birth and at 6 months of age.** Pearson's rho = 0.23, p-value = 0.03.

received vitamin D supplementation. Future determination and education on the optimum dose of prenatal and postnatal vitamin D intake or supplementation for the high-risk population in our setting may be warranted.

**Table 2. The change in vitamin D concentration between birth and 6 months of age.**

| Vitamin D status at birth[1] | Vitamin D status at 6 months of age[2] | | | | | Total |
|---|---|---|---|---|---|---|
| | >75 nmol/L | 50–75 nmol/L | 30–49.9 nmol/L | 12.5–29.9 nmol/L | <12.5 nmol/L | |
| >75 nmol/L | 3 (100%) | 0 (0%) | 0 (0%) | 0 (0%) | 0 (0%) | **3** |
| 50–75 nmol/L | 16 (67%) | 6 (25%) | 1 (4%) | 1 (4%) | 0 (0%) | **24** |
| 30–49.9 nmol/L | 51 (65%) | 19 (24%) | 6 (8%) | 2 (3%) | 0 (0%) | **78** |
| 12.5–29.9 nmol/L | 66 (52%) | 45 (35%) | 11 (9%) | 5 (4%) | 0 (0%) | **127** |
| <12.5 nmol/L | 4 (24%) | 7 (41%) | 3 (18%) | 2 (12%) | 1 (6%) | **17** |
| **Total** | **140 (56%)** | **77 (31%)** | **21 (8%)** | **10 (4%)** | **1 (1%)** | **249** |

[1] The 249 participants who had complete vitamin D measurement pairs (cord and 6-month venous blood) that included in the table.

[2] As per study protocol and due to the half-life of [25(OH)D] is 2–3 weeks, only venous blood collected timely within 6 months (± 4 weeks) were included.

[3] Light grey = deficiency at birth; dark grey = deficiency at birth and remained deficiency at 6 months of age.

**Table 3. Vitamin D-related factors (N = 350)[1].**

| Variables | Kota Yogyakarta (Urban, N = 127) | Kulon Progo (Rural, N = 223) | All (N = 350) |
|---|---|---|---|
| Exclusive Breastfeeding (EBF)–n (%), N = 333 | 76 (64%) | 118 (55%) | 194 (58%) |
| Maternal vitamin D supplementation in pregnancy–n (%) | 8 (6%) | 15 (7%) | 23 (7%) |
| Infant factors at six months of age: | | | |
| • Formula feeding–n (%), N = 334 | 26 (22%) | 52 (24%) | 78 (23%) |
| • Consumption vitamin D rich food at six months of age | | | |
| Egg yolk–n (%), N = 334 | 2 (2%) | 15 (7%) | 17 (5%) |
| Red meat–n (%), N = 334 | 2 (2%) | 1 (1%) | 3 (1%) |
| Fish–n (%), N = 334[3] | 5 (4%) | 8 (4%) | 13 (4%) |
| • Arm and legs uncovered more than half of the outdoor time on weekdays[2] –n (%), N = 333 | | | |
| Never or sometimes | 8 (7%) | 11 (5%) | 19 (6%) |
| Always | 112 (93%) | 202 (95%) | 314 (94%) |
| • Day time spent outdoors–n (%), N = 334 | | | |
| < 10 am | 97 (81%) | 183 (86%) | 280 (84%) |
| 10 am to 4 pm | 14 (12%) | 21 (10%) | 35 (10%) |
| > 4 pm | 9 (7%) | 10 (4%) | 19 (6%) |

[1] N = 350, unless otherwise specified.

[2] **Never or sometimes** defined as arm and legs were all covered or partially uncovered in more than half of the outdoor time at six months of age on weekdays.

[3] Fish served as freshwater fish such as catfish or salted fish.

Our finding of an increased risk of vitamin D deficiency in EBF infants is consistent with previous studies [37, 38]. Many studies from a range of settings have consistently reported that breast milk has poor vitamin D content [39, 40]. Even settings where lifetime sunlight exposure was abundant or maternal vitamin D deficiency was uncommon, EBF infants only received 20% of their daily vitamin D requirement ie of 400 IU [39, 41]. Given the many nutritional and health benefits of EBF in early infancy, especially in populations in resource-limited settings [42, 43], the implications of breast milk being a poor source of vitamin D intake for infants requires careful consideration. Our findings indicate that higher postnatal skin-sun exposure protects against vitamin D deficiency at six months of age in this group. Thus, adequate postnatal sunbathing may be particularly important to prevent vitamin D deficiency among exclusively breastfed infants.

International organizations such as UNICEF, national guidelines such as NICE and vitamin D experts have formulated recommendations for EBF practices in relation to ensuring the potential health benefits of vitamin, including vitamin D supplementation up to 400 IU for EBF infants or breastfeeding mothers [6, 30, 44, 45]. Experts agree that 50 nmol/L is the minimum 25-hydroxyvitamin D concentration for bone homeostasis. In addition to the prevention of rickets, there is some evidence that vitamin D protects against infections [2, 46, 47] and neurodevelopmental delay [48]. Vitamin D plays a direct role in regulating the expression of human antimicrobial peptides (cathelicidin and human beta-defensin 2) for pathogen elimination and reduction of viral replication [49]. However, findings from clinical studies are inconsistent and potentially biased by confounders with a lack of high-quality clinical trials, and so there is no consensus yet on minimum vitamin D requirements for "non-skeletal" benefits [1, 45, 50].

Some strategies have been proposed to prevent vitamin D deficiency in infants, but implementation is challenging. Indirect supplementation of breastfeeding women is impractical as

**Table 4. Maternal demographic profiles, sun-related phenotypes, and behaviors during pregnancy by cord blood vitamin D status at birth.**

| Variables | ≥50 nmol/L (N = 36) | <50 nmol/L (N = 308) | OR (95% CI) | p-value | Adjusted OR (95% CI) | p-value |
|---|---|---|---|---|---|---|
| | | | Cord blood vitamin D concentration | | | |
| | | | <50 vs ≥50 nmol/L | | | |
| **Maternal age[3], n = 344** | | | | | | |
| • ≥35 years old | 10 (28%) | 46 (15%) | Ref | | Ref | |
| • <35 years old | 26 (72%) | 262 (85%) | 2.19 (0.99–4.85) | 0.05 | 2.01 (0.86–4.72) | 0.11 |
| **Maternal education[3], n = 343** | | | | | | |
| • Middle school or less | 15 (43%) | 67 (22%) | Ref | | Ref | |
| • High school or university | 20 (57%) | 241 (78%) | **2.70 (1.31–5.55)** | **0.007** | **2.44 (1.09–5.45)** | **0.03** |
| **Maternal skin type[3], n = 320** | | | | | | |
| • Type II–white, fair | 1 (3%) | 9 (3%) | Ref | | Ref | |
| • Type III–medium, white to olive | 10 (30%) | 131 (46%) | 1.46 (0.17–12.67) | 0.73 | 1.54 (0.18–13.61) | 0.70 |
| • Type IV–olive, moderate brown | 17 (52%) | 128 (45%) | 0.84 (0.10–7.02) | 0.87 | 0.87 (0.10–7.40) | 0.90 |
| • Type V–Brown, dark brown | 5 (15%) | 19 (7%) | 0.42 (0.04–4.17) | 0.46 | 0.51 (0.05–5.11) | 0.56 |
| **Time spent outdoors during pregnancy in hours[4] (median, IQR)** | | | | | | |
| • Weekdays, n = 340 | | | | | | |
| 15 minutes or less | 0.25 (0.25–0.25) | 0.25 (0.25–0.25) | Ref | | Ref | |
| an hour or less | 0.5 (0.5–1) | 0.5 (0.5–1) | 0.17 (0.02–1.26) | 0.08 | 0.19 (0.02–1.48) | 0.11 |
| 2 hours | 2 (2–2) | 2 (2–2) | **0.11 (0.01–0.90)** | **0.039** | **0.10 (0.01–0.83)** | **0.03** |
| 3 hours | 4 (4–5) | 3 (3–4) | **0.08 (0.01–0.73)** | **0.025** | **0.09 (0.01–0.79)** | **0.03** |
| • Weekends, n = 317 | | | | | | |
| 15 minutes or less | 0.25 (0.25–0.25) | 0.25 (0.2–0.25) | Ref | | Ref | |
| an hour or less | 0.5 (0.5–1) | 1 (0.5–1) | 0.16 (0.02–1.19) | 0.07 | 0.18 (0.02–1.40) | 0.10 |
| 2 hours | 2 (2–2) | 2 (2–2) | 0.13 (0.02–1.08) | 0.06 | 0.12 (0.01–1.03) | 0.05 |
| 3 hours | 4 (4–5) | 3 (3–4) | **0.10 (0.01–0.83)** | **0.033** | **0.10 (0.01–0.90)** | **0.04** |
| **Lower skin-sun exposure score[1,5]** | | | | | | |
| • Weekdays, n = 340 | 5 (4–6) | 5 (3–6) | 0.98 (0.83–1.16) | 0.83 | 0.98 (0.82–1.17) | 0.81 |
| • Weekend, n = 317 | 5 (4–6) | 5 (2–6) | 1.00 (0.85–1.18) | 0.97 | 1.00 (0.83–1.19) | 0.96 |
| **Cumulative composite estimate UVR exposure[2] (per 100 mW/m²/nm/hour),** | | | | | | |
| • Weekdays, n = 316 | 3.63 (1.92–7.44) | 3.47 (1.76–6.88) | **0.92 (0.85–0.99)** | **0.030** | **0.91 (0.85–0.99)** | **0.02** |
| • Weekend, n = 339 | 3.68 (1.92–7.47) | 3.44 (1.76–3.84) | **0.90 (0.84–0.97)** | **0.008** | **0.90 (0.83–0.97)** | **0.007** |
| Maternal vitamin D supplementation during pregnancy[3,6] | 5 (14%) | 17 (5%) | 0.36 (0.13–1.05) | 0.06 | **0.15 (0.04–0.50)** | **0.002** |

[1] **Skin-sun exposure score:** the frequency of uncovered/exposed arms and legs (1 = never, 2 = less than 50% of the time, 3 = more than 50% of the time, 4 = all the time). The score for skin exposure was the summation of each category (2 = min. sun exposure, 8 = max. sun exposure).

[2] **Cumulative composite estimate UVR exposure:** the product of time spent outdoors multiplied by the ambient UVB during the period of time.

[3] Adjusted for sex, socio-economic status, the month of birth and mother's skin types.

[4] Adjusted for sex, socio-economic status, the month of birth, skin-sun exposure score, and mother's skin types.

[5] Adjusted for sex, socio-economic status, the month of birth, time spent outdoor and mother's skin types.

[6] Adjusted for sex, socio-economic status, the month of birth, mother's skin types, and cumulative composite estimate UVR exposure in weekdays.

[7] Multivitamin containing vitamin D.

the vitamin D content in supplemented breast milk does not always provide the infants with the amount currently recommended of 400 IU per day. To achieve this amount, high doses are often required of 4000–6400 IU per day with a potential increased risk of hypercalcemia and nephrocalcinosis [30, 44, 45, 51]. Therefore, direct infant supplementation is now preferentially recommended [30, 44, 45, 51]. In Indonesia, affordability is an issue as currently available

**Table 5. Sun-related phenotypes, behavior, and UVR in the first 6-month of infancy by vitamin D status at 6 months of age.**

| Variables | Serum vitamin D concentration at 6 months of age | | | | | |
|---|---|---|---|---|---|---|
| | ≥50 nmol/L (N = 222) | <50 nmol/L (N = 33) | <50 vs ≥50 nmol/L | | | |
| | | | OR (95% CI) | p-value | Adjusted OR (95% CI) | p-value |
| **Cord blood vitamin D concentration,[1] n = 249** | | | | | | |
| • >50 nmol/L | 25 (12%) | 2 (6%) | Ref | | Ref | |
| • 30–49 nmol/L | 70 (32%) | 8 (25%) | 1.43 (0.28–7.19) | 0.67 | 1.36 0.26–7.23) | 0.72 |
| • 12.5–29 nmol/L | 111 (51%) | 16 (50%) | 1.80 (0.39–8.34) | 0.45 | 2.06 (0.41–10.30) | 0.38 |
| • <12.5 nmol/L | 11 (5%) | 6 (19%) | **6.82 (1.18–39.25)** | **0.032** | **7.73 (1.20–49.60)** | **0.03** |
| **Exclusive breastfeeding[2], n = 255** | | | | | | |
| • No | 93 (42%) | 7 (21%) | Ref | | Ref | |
| • Yes | 129 (58%) | 26 (79%) | **2.68 (1.12–6.43)** | **0.028** | **2.64 (1.07–6.49)** | **0.04** |
| **Infants Skin type[3], n = 253** | | | | | | |
| • Type II–white, fair | 39 (18%) | 12 (36%) | Ref | | Ref | |
| • Type III–medium, white to olive | 141 (64%) | 18 (55%) | **0.42 (0.18–0.93)** | **0.034** | **0.40 (0.17–0.93)** | **0.03** |
| • Type IV–olive, moderate brown | 40 (18%) | 3 (9%) | **0.24 (0.06–0.93)** | **0.039** | **0.23 (0.06–0.92)** | **0.04** |
| **Nutritional status at 6 months (weight for length at 6 months of age)[4], n = 157** | | | | | | |
| • Overweight/ obese (>2 SD) | 6 (4%) | 1 (5%) | 1.34 (0.15–11.94) | 0.79 | 2.70 (0.26–28.24) | 0.41 |
| • Normal (≥- 2 SD to ≤2 SD) | 121 (88%) | 15 (75%) | Ref | | n/a | |
| • Wasted (≥- 3 SD to <-2 SD) | 8 (6%) | 4 (20%) | **4.03 (1.08–15.21)** | **0.038** | 3.48 (0.79–15.41) | 0.10 |
| • Severely wasted (<-3 SD) | 2 (2%) | 0 (0%) | n/a | | n/a | |
| **People sleeping in the same room with infant[4], n = 254** | | | | | | |
| • 2 people or less | 171 (77%) | 14 (44%) | Ref | | Ref | |
| • >2 people | 51 (23%) | 18 (56%) | **4.31 (2.01–9.27)** | **<0.001** | **4.96 (2.18–11.30)** | **<0.001** |
| **Infancy sun-related behavior** | | | | | | |
| **Lower skin-sun exposure score (cumulative score in 6 months, min 12, max 48)[3] –median (IQR)** | | | | | | |
| • Weekdays, n = 248 | 42 (40–46) | 40 (36–44) | **1.10 (1.03–1.18)** | **0.008** | **1.12 (1.04–1.20)** | **0.004** |
| • Weekends, n = 227 | 42 (40–46) | 42 (36–46) | 1.08 (0.99–1.16) | 0.06 | **1.09 (1.00–1.19)** | **0.03** |
| **Time spent outdoors in hours (the average of in the last 6 months)[4]** | | | | | | |
| • Weekdays, n = 250 | | | | | | |
| an hour or less | 174 (80%) | 26 (81%) | Ref | | Ref | |
| 1–2 hours | 39 (18%) | 5 (16%) | 0.86 (0.31–2.38) | 0.77 | 0.82 (0.28–2.37) | 0.71 |
| 2 hours or more | 5 (2%) | 1 (3%) | 1.34 (0.15–11.91) | 0.79 | 2.53 (0.26–24.40) | 0.42 |
| • Weekends, n = 224 | | | | | | |
| an hour or less | 153 (76%) | 21 (78%) | Ref | | Ref | |
| 1–2 hours | 44 (22%) | 6 (22%) | 0.99 (0.38–2.61) | 0.99 | 0.92 (0.34–2.50) | 0.86 |
| 2 hours or more | 4 (2%) | 0 (0%) | n/a | | n/a | |
| **Cumulative composite estimate UVR exposure[2] (per 100 mW/m²/nm/hour)–median (IQR)** | | | | | | |
| • Weekdays, n = 249 | 3.80 (2.94–5.10) | 3.76 (2.50–4.58) | 0.98 (0.81–1.17) | 0.81 | 0.99 (0.82–1.20) | 0.91 |
| • Weekends, n = 228 | 3.86 (2.94–5.13) | 3.73 (2.74–4.94) | 1.00 (0.84–1.20) | 0.97 | 1.04 (0.86–1.27) | 0.68 |
| Formula feeding at 6 months of age[2] | 50 (23%) | 2 (6%) | **0.22 (0.05–0.96)** | **0.04** | **0.20 (0.05–0.90)** | **0.04** |

**Skin-sun exposure score:** the frequency of uncovered/exposed arms and legs (1 = never, 2 = less than 50% of the time, 3 = more than 50% of the time, 4 = all the time). The score for skin exposure was the summation of each category (2 = min. sun exposure, 8 = max. sun exposure), The cumulative score for skin exposure in 6 months was minimum 12 and maximum 48.

**Cumulative composite estimate UVR exposure:** the product of time spent outdoors multiplied by the ambient UVB during the period of time.

n/a: not applicable.

[1] adjusted for sex and family income, EBF, skin types, the month of birth and the month of blood collection.

[2] adjusted for sex, family income, skin types and the month of blood collection.

[3] adjusted for sex, family income, EBF, time spent outdoors, and the month of blood collection.

[4] adjusted for sex, family income, EBF, cumulative skin-sun exposure, skin type and the month of blood collection.

formulations (400 IU/5 mL) of vitamin D for infants are expensive, not covered by universal health coverage and so are beyond the reach of most low-income families. Our findings indicate the role of 'safe' sun exposure for the mother in pregnancy, the infant postnatally, and particularly among EBF infants requires further consideration. These findings should be taken into account for recommendations aimed to avoid early life vitamin D deficiency.

The current study has several strengths. We performed gold standard vitamin D assay (LC-MS/MS) and measured a broad range of sun-related variables such as sun-phenotypes, sun behavior, environmental factors, infant feeding practice, and dietary intake in both mothers during pregnancy (retrospectively) and infants (for 12 months, prospectively). A prospective design allowed us to measure sun behavior cumulatively for six months prior to venous blood collection. The infant sun exposure measures used have previously been shown to related to vitamin D status [21]. The current study has some potential limitations. The measurement of sun exposure behavior was based on parental recall at four-weekly intervals that are not as accurate as a daily diary record for sun exposure time or by using a device such as a dosimeter to record the actual sun exposure time [13]. Maternal vitamin D status was not measured to examine the correlation with the concentration of cord blood vitamin D. Vitamin D deficiency has been reported in 20% of 140 Indonesian women in the first trimester of pregnancy but Indonesian data that correlate vitamin D status during pregnancy with cord blood are unavailable [48].

## Conclusions

The high prevalence of vitamin D deficiency at birth in Indonesian infants had improved in the majority of infants by six months of age most likely due to the current cultural practice of infant sunbathing. Risk factors for vitamin D deficiency at six months of age were less infant skin area exposed to direct sunlight, severe vitamin D deficiency at birth, and EBF until six months of age. Among EBF infants, higher skin-sun exposure mitigated vitamin D deficiency risk. The role of 'safe' morning sun exposure in infants and mothers to prevent early life vitamin D deficiency in populations with medium to dark brown skin pigmentation requires further consideration. Effective interventions to improve vitamin D status in EBF infants that are locally appropriate need further consideration and evaluation.

## Acknowledgments

We thank all participants and parents who participated in this study, all study site staffs who helped with the study recruitment. Lastly, we especially thank all IPADS research assistants who assisted with data collection and study conduct.

## Author Contributions

**Conceptualization:** Vicka Oktaria, Stephen M. Graham, Rina Triasih, Yati Soenarto, Julie E. Bines, Anne-Louise Ponsonby, Margaret Danchin.

**Data curation:** Vicka Oktaria.

**Formal analysis:** Vicka Oktaria, Michael W. Clarke, Hera Nirwati.

**Funding acquisition:** Vicka Oktaria, Stephen M. Graham, Yati Soenarto, Margaret Danchin.

**Investigation:** Vicka Oktaria, Stephen M. Graham, Rina Triasih, Yati Soenarto, Julie E. Bines, Rizka Dinari, Margaret Danchin.

**Methodology:** Vicka Oktaria, Stephen M. Graham, Rina Triasih, Julie E. Bines, Anne-Louise Ponsonby, Margaret Danchin.

**Project administration:** Rizka Dinari, Hera Nirwati.

**Resources:** Vicka Oktaria, Stephen M. Graham, Yati Soenarto, Margaret Danchin.

**Supervision:** Vicka Oktaria, Stephen M. Graham, Rina Triasih, Yati Soenarto, Julie E. Bines, Anne-Louise Ponsonby, Margaret Danchin.

**Validation:** Vicka Oktaria, Stephen M. Graham, Michael W. Clarke, Rizka Dinari, Margaret Danchin.

**Writing – original draft:** Vicka Oktaria.

**Writing – review & editing:** Vicka Oktaria, Stephen M. Graham, Rina Triasih, Yati Soenarto, Julie E. Bines, Anne-Louise Ponsonby, Michael W. Clarke, Rizka Dinari, Hera Nirwati, Margaret Danchin.

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
