## [Decision Letter · Decision Letter 0]

14 Jul 2020

PONE-D-20-14874

The prevalence and determinants of vitamin D deficiency in Indonesian infants at birth and six months of age

PLOS ONE

Dear Dr. oktaria,

Thank you for submitting your manuscript to PLOS ONE. After careful consideration, we feel that it has merit but does not fully meet PLOS ONE’s publication criteria as it currently stands. Therefore, we invite you to submit a revised version of the manuscript that addresses the points raised during the review process.

We look forward to receiving your revised manuscript.

Kind regards,

Michal Zmijewski

Academic Editor

PLOS ONE

Journal Requirements:

2.Thank you for stating the following in the Acknowledgments Section of your manuscript:

'V.O. holds scholarship from Indonesia Endowment Fund for Education (LPDP), Ministry of Finance, Indonesia. M.D. holds the David Bickart Clinician Researcher Fellowship from the University of Melbourne. M.W.C. is affiliated to Metabolomics Australia, University of Western Australia, Perth, Western Australia, Australia. This was supported by infrastructure funding from the Western Australian State Government in partnership with the Australian Federal Government, through Bioplatforms Australia and the National Collaborative Research Infrastructure Strategy (NCRIS).'

'VO. Schlumberger foundation faculty for the future. https://www.fftf.slb.com.

SMG. Murdoch Childrens Research Institute internal theme grants 2014 and 2016. https://www.mcri.edu.au

The funders had no role in study design, data collection and analysis, decision to publish, or preparation of the manuscript'

Additional Editor Comments:

Please focus introduction on the main subject of study

and explain the vitamin D supplementation in infants in idetails.

Please, follow the instruction from reviewers.

Reviewers' comments:

Reviewer's Responses to Questions

**Comments to the Author**

1. Is the manuscript technically sound, and do the data support the conclusions?

Reviewer #1: Partly

Reviewer #2: Yes

2. Has the statistical analysis been performed appropriately and rigorously? 

Reviewer #1: Yes

Reviewer #2: Yes

3. Have the authors made all data underlying the findings in their manuscript fully available?

Reviewer #1: Yes

Reviewer #2: Yes

4. Is the manuscript presented in an intelligible fashion and written in standard English?

Reviewer #1: No

Reviewer #2: Yes

5. Review Comments to the Author

Reviewer #1: -The study topic is novel and practical, however there are many confounding factors that are not considered in the study.

-Supplementation of vitamin D in infants has not been considered thoroughly.

-The introduction should be summarized.

-The interpretation of findings should be expanded.

-The English writing should be improved.

Reviewer #2: This is an interesting paper and valuable data on vitamin D status in infancy in a tropical setting.

I have some suggestions for enhancement of this manuscript as follows.

What data on infant feeding practices was collected, please provide more details, especially regarding the use of infant formula? This is essential information given that infant formula is supplemented with vitamin D.

Please provide the latitude range of the study sites and/or participant locations.

Given that the focus of the introduction of this manuscript was on infectious diseases, why was there no infectious diseases or any other clinical outcome data included? If this is not to be included, then this manuscript needs an introduction re-write to introduce the content of this manuscript with more relevance.

Lines 54-55 (Abstract): Delete 95% CI 86 - 93% and 95% CI 9-18% as not needed.

Lines 124-156: I would recommend that the Vitamin D assay method and definitions of vitamin D deficiency lines 144-147 are combined, but move lines 134-137 to be combined with lines 147-156 in a separate section headed Ultraviolet light exposure.

Line 181: (n= 60)? Should this be n=6 if 344 cord blood samples were collected as stated in line 173?

Lines 180-185: This Figure 1 description is confusing. A more traditional cohort study participant flow diagram should be included instead.

Line 190: Delete 95% CI: 86 - 93% as this doesn’t make sense why this needs to be included.

Line 191: Delete 95% CI 9-18% as this doesn’t make sense why this needs to be included.

Table 1 on participant demographics contains results like exclusive breastfeeding rates, arm and legs uncovered more than half of the outdoor time at six months of age and day time spent outdoor at six months of age?

Table 1 needs to include season/month of birth.

Figure 1: First box should state - 350 participants recruited rather than 350 participants with available blood samples.

6. PLOS authors have the option to publish the peer review history of their article (what does this mean?). If published, this will include your full peer review and any attached files.

Reviewer #1: **Yes: **Roya Kelishadi

Reviewer #2: No

---

## [Author Response · Author response to Decision Letter 0]

4 Aug 2020

The prevalence and determinants of vitamin D deficiency in Indonesian infants at birth and six months of age 

Submission ID: PONE-D-20-14874 

Response to reviewers: 

Editor

Please focus introduction on the main subject of study and explain the vitamin D supplementation in infants in details

Author response:

We have amended the introduction as suggested and added some information related to vitamin D supplementation in the text.

Change made in the manuscript:

“Over recent decades, there has been a rapid growth of research interest in perinatal vitamin D deficiency due to a range of potential roles of vitamin D for prevention of disease, such as low birth weight in newborns or infections in early childhood [1-5]. A temperate country such as Australia with a high prevalence of infant vitamin D deficiency, particularly in winter, has implemented preventive measures through routine vitamin D supplementation for high-risk pregnant women and infants (i.e. those with dark skin type and covering clothing), in addition to guidance around safe sun exposure practices [6]. Despite abundant sunlight, vitamin D deficiency is also common in tropical countries, but under-diagnosed. Recent studies have reported that one in every two African and nine in every ten Thai neonates were vitamin D deficient [7, 8]. In tropical regions that experience high neonatal and child health morbidity and mortality, such as in Africa and Southeast Asia [9, 10], sustaining adequate vitamin D level could be a low-cost public health measure to reduce the burden of infectious diseases. However, efforts to prevent, detect, and treat vitamin D deficiency in many of these regions are minimal or absent. 

Understanding the epidemiology and identification of risk factors for vitamin D deficiency in early life is important for future development of preventive strategies but they are not currently well understood in Indonesia. Indonesia is the most populous country in Southeast Asia and has the fifth-highest global burden of infant morbidity and mortality from acute respiratory infections [11, 12].” (Page 4-5, line 83 – 135)

Author response:

Routine vitamin D supplementation in healthy infants was not standard practice during our study and none of our participants received vitamin D supplementation.

Change made in the manuscript:

We have stated this in the discussion section

“Prenatal and postnatal vitamin D supplementation is not routinely recommended in our setting. Further, there are limited selections of food with a high content of vitamin D in the daily intake of Indonesian people [36]. Only 5% of our study infants had egg yolk in the first six months, mainly those who were living in the rural area, and even less had had fish or red meats. A minority (7%) of mothers reported taking a multivitamin supplement that contained vitamin D during pregnancy with 50% reporting a daily vitamin D intake below 400 IU, the minimum recommended daily dose in pregnancy [6]. None of the participating infants received vitamin D supplementation. Future determination and education on the optimum dose of prenatal and postnatal vitamin D intake or supplementation for the high-risk population in our setting may be warranted.” (Page 25-26, line 399-408)

Author response:

We have added some information on vitamin D-containing food such as infant formula and dietary intake such as egg yolks, red meat and fish

Change made in the manuscript:

As presented in Table 3 (Table 3 page 14-15)

Reviewer 1

Comment 1: Supplementation of vitamin D in infants has not been considered thoroughly

Author response:

During the study, vitamin D supplementation in healthy infants was not routine practice. Vitamin D is expensive and usually only prescribed by a pediatrician under certain medical conditions. None of our participating infants had vitamin D supplementation, but we did collect data on consumption of vitamin D rich foods such as egg yolk, red meat, fish and infant formula. This information is now presented in Table 3

Change made in the manuscript:

We have stated this in the discussion section

“Prenatal and postnatal vitamin D supplementation is not routinely recommended in our setting. Further, there are limited selections of food with a high content of vitamin D in the daily intake of Indonesian people [36]. Only 5% of our study infants had egg yolk in the first six months, mainly those who were living in the rural area, and even less had had fish or red meats. A minority (7%) of mothers reported taking a multivitamin supplement that contained vitamin D during pregnancy with 50% reporting a daily vitamin D intake below 400 IU, the minimum recommended daily dose in pregnancy [6]. None of the participating infants received vitamin D supplementation. Future determination and education on the optimum dose of prenatal and postnatal vitamin D intake or supplementation for the high-risk population in our setting may be warranted.” (Page 25-26, line 399-408)

Data on some food that may contain vitamin D are presented in Table 3 (Table 3 page 14-15)

Comment 2: The introduction should be summarized

Author response:

We have amended the introduction

Change made in the manuscript:

As written in the Introduction section (Page 4-5, line 83 – 135)

Comment 3: The interpretation of findings should be expanded

Author response: 

We have expanded the findings and study interpretation

Change made in the manuscript:

We have added discussion on maternal intake of a multivitamin formulation containing vitamin D. (Page 25-26, line 399-408)

Comment 4: The English writing should be improved

Author response: 

Thank you very much for your feedback. This manuscript has undergone additional review by our English native authors prior to resubmission. The quality of English language is improved.

Reviewer 2

Comment 1: What data on infant feeding practices was collected, please provide more details, especially regarding the use of infant formula? This is essential information given that infant formula is supplemented with vitamin D.

Author response: 

We collected data on infant formula and some foods such as egg yolk, red meat and fish that contain vitamin D.

This information has now been presented in Table 3

Change made in the manuscript:

This additional data is presented in Table 3 (Table 3 page 14-15)

Comment 2: Please provide the latitude range of the study sites and/or participant locations

Author response: 

We have now added the latitude for the study locations

Change made in the manuscript:

“This study was part of the Indonesian Pneumonia and Vitamin D status (IPAD study), a community-based birth cohort study conducted between December 2015 to December 2017 in nine Primary Healthcare Centers and five private clinics in Kota Yogyakarta (7.80� S, 110.4� E) and Kulon Progo regencies (7.83� S, 110.2� E) , Yogyakarta province, Indonesia.” (Page 5-6, line 148 – 159)

Comment 2: Given that the focus of the introduction of this manuscript was on infectious diseases, why was there no infectious diseases or any other clinical outcome data included? If this is not to be included, then this manuscript needs an introduction re-write to introduce the content of this manuscript with more relevance.

Author response: 

Thank you for your comment. The infectious disease data (acute respiratory infections) is presented in a separate publication and is not the focus of this manuscript.

We have now rewritten the introduction to focus more on vitamin D prevalence and risk factors (Page 4-5, line 83 – 146)

Comment 3: Lines 54-55 (Abstract): Delete 95% CI 86 - 93% and 95% CI 9-18% as not needed.

Author response: 

We have deleted the 95% CIs.

Change made in the manuscript:

“Between December 2015 to December 2017, 350 maternal-newborn participants were recruited and followed up. Vitamin D deficiency was detected in 90% (308/344) of cord blood samples and 13% (33/255) of venous blood samples at six months.” (Page 3, Line 54 – 55)

Comment 4: Lines 124-156: I would recommend that the Vitamin D assay method and definitions of vitamin D deficiency lines 144-147 are combined but move lines 134-137 to be combined with lines 147-156 in a separate section headed Ultraviolet light exposure.

Author response:

We have combined the vitamin D assay and definition into one section and have now included a separate section that addresses Ultraviolet exposure

Change made in the manuscript:

“Study assays and measurement for vitamin D 

We measured total serum 25-hydroxyvitamin D3 and 25-hydroxyvitamin D2 concentrations using liquid chromatography tandem mass spectrometry (LC-MS/MS), the current gold standard for vitamin D measurement. The LC-MS/MS analysis was performed by Metabolomics Australia, at the University of Western Australia. The analytical sensitivity (or limit of detection) and the functional sensitivity (or limit of quantitation) of the assay for 25-hydroxyvitamin D3 is 0.5 nM and 2 nmol/L respectively. The center has been certified by the Center of Disease Control and Prevention for standardized vitamin D measurement (r2 =0.99) [18]. Vitamin D deficiency was defined as a concentration of 25-hydroxyvitamin D < 50 nmol/L. The vitamin D concentrations at birth and at six months of age were also categorized by severity using the categories suggested by Paxton et al.: mild deficiency (30-49 nmol/L); moderate deficiency (12.5-29 nmol/L) and severe deficiency (< 12.5 nmol/L).[19]”

And 

“Data acquisition for Ultraviolet B 

Ultraviolet B (UVB) and Erythema Daily Dose Rate data for the Yogyakarta Special Region were downloaded monthly and obtained from the Aura Data Validation Centre as a part of the National Aeronautics and Space Administration, Goddard Space Flight Centre, Greenbelt, Maryland. (URL link: https://avdc.gsfc.nasa.gov/index.php?site=2057856112&id=79)” (Page 6-7, Line 174 – 194)

Comment 5: Line 181: (n= 60)? Should this be n=6 if 344 cord blood samples were collected as stated in line 173?

Author response:

Thank you for this correction. We have now revised Figure 1 as attached (Figure 1 attachment)

Change made in the manuscript:

As displayed in the new Figure 1

Comment 6: Lines 180-185: This Figure 1 description is confusing. A more traditional cohort study participant flow diagram should be included instead.

Author response:

We have now revised Figure 1 as attached

Change made in the manuscript:

As displayed in the new Figure 1 (Figure 1 attachment)

Comment 7: Line 190: Delete 95% CI: 86 - 93% as this doesn’t make sense why this needs to be included.

Author response:

We have deleted the 95% CI

Change made in the manuscript:

“The prevalence of vitamin D deficiency was 90% in 344 participants at birth and 13% in 255 participants at six months (� 4 weeks) (Fig 2).” (Page 12, line 265-266 – 272)

Comment 8: 

Line 191: Delete 95% CI 9-18% as this doesn’t make sense why this needs to be included

Author response:

We have deleted the 95% CI

Change made in the manuscript:

“The prevalence of vitamin D deficiency was 90% in 344 participants at birth and 13% in 255 participants at six months (� 4 weeks) (Fig 2).” (Page 12, line 265-266)

Comment 9: Table 1 on participant demographics contains results like exclusive breastfeeding rates, arm and legs uncovered more than half of the outdoor time at six months of age and daytime spent outdoor at six months of age?

Author response:

Thank you for your feedback.

The information on exclusive breastfeeding rates and sun behavioral has been removed from Table 1 and presented in a separate table (Table 3 – Vitamin D-related factors)

Change made in the manuscript:

Removed from Table 1 (Table 1 - Page 10-11)

New data listed in Table 3 (Table 3-Page 14-15)

Comment 10: Table 1 needs to include season/month of birth

Author response:

We have added month of birth in Table 1

Change made in the manuscript:

Month of birth and season has been listed in Table 1. 

We have also added information in the table footnote “Indonesia is a tropical country with two seasons: dry season (around April to October) and wet season (around November to March)” (Table 1 - Page 10-11)

Comment 11: Figure 1: First box should state - 350 participants recruited rather than 350 participants with available blood samples

Author response:

We have now revised Figure 1 as attached

Change made in the manuscript:

As displayed in the new Figure 1 (Figure 1 attachment)

---

## [Decision Letter · Decision Letter 1]

26 Aug 2020

PONE-D-20-14874R1

The prevalence and determinants of vitamin D deficiency in Indonesian infants at birth and six months of age

PLOS ONE

Dear Dr. oktaria,

Thank you for submitting your manuscript to PLOS ONE. After careful consideration, we feel that it has merit but does not fully meet PLOS ONE’s publication criteria as it currently stands. Therefore, we invite you to submit a revised version of the manuscript that addresses the points raised during the review process.

We look forward to receiving your revised manuscript.

Kind regards,

Michal Zmijewski

Academic Editor

PLOS ONE

Additional Editor Comments (if provided):

Please, rewrite a little introduction as suggested by Reviewer 1.

Reviewers' comments:

Reviewer's Responses to Questions

**Comments to the Author**

1. If the authors have adequately addressed your comments raised in a previous round of review and you feel that this manuscript is now acceptable for publication, you may indicate that here to bypass the “Comments to the Author” section, enter your conflict of interest statement in the “Confidential to Editor” section, and submit your "Accept" recommendation.

Reviewer #1: All comments have been addressed

Reviewer #2: All comments have been addressed

2. Is the manuscript technically sound, and do the data support the conclusions?

Reviewer #1: Partly

Reviewer #2: (No Response)

3. Has the statistical analysis been performed appropriately and rigorously? 

Reviewer #1: Yes

Reviewer #2: (No Response)

4. Have the authors made all data underlying the findings in their manuscript fully available?

Reviewer #1: Yes

Reviewer #2: (No Response)

5. Is the manuscript presented in an intelligible fashion and written in standard English?

Reviewer #1: Yes

Reviewer #2: (No Response)

6. Review Comments to the Author

Reviewer #1: All comments are successfully considered, however I suggest that the conclusion section of the abstract would be re-written to be more concise and precise with a message for international readers.

Reviewer #2: (No Response)

7. PLOS authors have the option to publish the peer review history of their article (what does this mean?). If published, this will include your full peer review and any attached files.

Reviewer #1: **Yes: **Roya Kelishadi

Reviewer #2: No

---

## [Author Response · Author response to Decision Letter 1]

8 Sep 2020

The prevalence and determinants of vitamin D deficiency in Indonesian infants at birth and six months of age 

Submission ID: PONE-D-20-14874 

Response to reviewers: 

Editor

Please, rewrite a little introduction as suggested by Reviewer 1.

Further suggestions (email correspondence on 3 September 2020)

Rewrite a little just conclusion of the abstract and to include additional factors which might make this more international like effects of latitude or skin pigmentation (phototype), which could be of interest and the subject of further studies. Please also note that in many countries sun exposure of infants is not advices.

Author response:

Thank you very much for your suggestions. We have amended the conclusion in the abstract as suggested and added a few more details in the conclusion in the main body of the paper.

Change made in the manuscript:

In equatorial regions, the role of ‘safe’ morning sun exposure in infants and mothers in populations with medium to dark brown skin pigmentation and effective interventions to prevent vitamin D deficiency in newborns and EBF infants, need further consideration and evaluation.(Abstract’s conclusion section, page 4, line 66 – 69)

And

The role of ‘safe’ morning sun exposure in infants and mothers to prevent early life vitamin D deficiency in populations with medium to dark brown skin pigmentation requires further consideration. (Main body’s conclusion section, page 28, line 379-381)

Reviewer 1

All comments are successfully considered; however, I suggest that the conclusion section of the abstract would be re-written to be more concise and precise with a message for international readers.

Author response:

Thank you for your suggestion. We have amended the abstract’s conclusion as suggested and added a few more details in the main body’s conclusion.

Change made in the manuscript:

In equatorial regions, the role of ‘safe’ morning sun exposure in infants and mothers in populations with medium to dark brown skin pigmentation and effective interventions to prevent vitamin D deficiency in newborns and EBF infants, need further consideration and evaluation.(Abstract’s conclusion section, page 4, line 66 – 69)

And

The role of ‘safe’ morning sun exposure in infants and mothers to prevent early life vitamin D deficiency in populations with medium to dark brown skin pigmentation requires further consideration. (Main body’s conclusion section, page 28, line 379-381)

---

## [Editor Report · Decision Letter 2]

10 Sep 2020

The prevalence and determinants of vitamin D deficiency in Indonesian infants at birth and six months of age

PONE-D-20-14874R2

Dear Dr. oktaria,

We’re pleased to inform you that your manuscript has been judged scientifically suitable for publication and will be formally accepted for publication once it meets all outstanding technical requirements.

Kind regards,

Michal Zmijewski

Academic Editor

PLOS ONE

Additional Editor Comments (optional):

Thank you
---

## [Editor Report · Acceptance letter]

22 Sep 2020

PONE-D-20-14874R2 

The prevalence and determinants of vitamin D deficiency in Indonesian infants at birth and six months of age 

Dear Dr. Oktaria:

I'm pleased to inform you that your manuscript has been deemed suitable for publication in PLOS ONE. Congratulations! Your manuscript is now with our production department. 

Kind regards, 

on behalf of

Dr. Michal Zmijewski 

Academic Editor

PLOS ONE